# Preliminary Study on the Host Response to Bivalent and Monovalent Autogenous Vaccines against *Mycoplasma agalactiae* in Dairy Sheep

**DOI:** 10.3390/vetsci9120651

**Published:** 2022-11-22

**Authors:** Hany A. Hussein, Marco Tolone, Lucia Condorelli, Paola Galluzzo, Roberto Puleio, Irene Vazzana, Maria Luisa Scatassa, Gavino Marogna, Santino Barreca, Guido Ruggero Loria, Lucia Galuppo, Sergio Migliore

**Affiliations:** 1OIE Reference Laboratory for Contagious Agalactia, Istituto Zooprofilattico Sperimentale della Sicilia “A. Mirri”, 90129 Palermo, Italy; 2Dipartimento di Scienze Agrarie, Alimentari e Forestali, University of Palermo, 90128 Palermo, Italy; 3Area Igiene delle Produzioni Zootecniche e Benessere Animale, Istituto Zooprofilattico Sperimentale della Sicilia “A. Mirri”, 90129 Palermo, Italy; 4Centro Latte e Lotta Alle Mastiti, Istituto Zooprofilattico Sperimentale della Sicilia “A. Mirri”, 90129 Palermo, Italy; 5Istituto Zooprofilattico Sperimentale della Sardegna “G. Pegreffi”, 7100 Sassari, Italy; 6Azienda Sanitaria Provinciale 6, 90100 Palermo, Italy

**Keywords:** contagious agalactia, vaccines, *S. aureus*, *M. agalactiae*, dairy sheep, humoral immunity, somatic cell count

## Abstract

**Simple Summary:**

Contagious agalactia (CA), caused by *Mycoplasma agalactiae* (*M. agalactiae*), is a very important disease in dairy sheep. It is distributed worldwide, and causes a marked drop or even complete loss of milk production, arthritis, keratoconjunctivitis, pneumonia, and septicemia. The current study represents the first investigation evaluating the synergistic effect of combining *Staphylococcus aureus* (*S. aureus*) within the *M. agalactiae* vaccine. Apart from the previously known benefits of combined vaccines, such as reduced production costs, time and effort saving, simplifying the immunization schedule, and minimizing the stress of multiple vaccinations, the current study recorded for the first time that the involvement of *S. aureus* in inactivated vaccines against *M. agalactiae* elicited a higher antibody response against *M. agalactiae*, detected by ELISA, than the monovalent *M. agalactiae* vaccine. The humoral immune response and hematological parameters were significantly improved and manifested by the increased antibody titre, total leukocytic count, neutrophils, and blood platelets. Moreover, the local glandular immunity of the udder, assessed by the somatic cell count (SCC), was enhanced.

**Abstract:**

In Italy, dairy sheep farming represents a vital agro-industry sector, but it is still challenged by contagious agalactia (CA), which is endemic there, and vaccination is the most economical and sustainable tool for control. This study aimed to evaluate the combined *Mycoplasma agalactiae (Ma)-Staphylococcus aureus* (*Sa*) vaccine *(Ma–Sa*) against the *Ma* monovalent vaccine in ewes. Twelve primiparous *Ma*-free ewes were randomly grouped into three equal groups: first, the control group injected with placebo, second, the group vaccinated with the *Ma* monovalent vaccine, and third, the group vaccinated with *Ma–Sa* combined vaccine, with two S/C doses at 45-day intervals. The animals were examined for serological, hematological, and somatic cell count (SCC) changes for 17 successive weeks. A significant increase in anti-*Ma* antibody mean titers, leukocytes, and platelets was observed in the vaccinated animals, with the highest values in those who received the combined vaccine. Neutrophils were high only in the animals who received the combined vaccine. SCC was lower in the vaccinated animals during the first six weeks. This study concludes that the combined *Ma–Sa* vaccines enhance immune response and potentiate its efficacy against *Ma*. This improvement might be attributed to the sensitization/activation effect of *S. aureus* on platelets, which are recoded to act as a key regulator for the coordination of all components of the innate immune system. Even though this study included a small number of animals, its findings about the potentialities of this inactivated vaccine in the control of CA are strongly encouraging. Further confirmation might be needed through additional replicates and a challenge study is needed before proceeding with widespread use.

## 1. Introduction

Global sheep and goat milk production has more than doubled during the last decades [1]. In Italy, sheep husbandry has become one of the most important sectors of the agro-industry, as confirmed by the constant and significant increase in sheep raised annually [2].

Contagious agalactia (CA) is an important disease of small ruminants, causing a marked drop or even complete loss of milk production. According to the World Organization for Animal Health (WOAH, previously known as OIE), CA is a notifiable disease [3] and is geographically widespread, particularly in countries with developed dairy sheep farming, and is enzootic in Southern Europe, the Middle East, Asia, and North Africa [4,5]. It has recently become even more important because of the worldwide increase in small ruminant milk production [6]. In Italy, the disease was firstly reported in 1816 [7].

Clinical outcomes of CA infection are not restricted to mammary glands, and are gathered under the acronym MAKePS, which stands for mastitis, arthritis, keratoconjunctivitis, pneumonia, and septicemia [8]. It can also cause abscesses in the udder and enlargement of the retromammary lymph nodes in all ages of animal, and abortions in pregnant females [9,10]. Economic losses induced by CA are due to reduced milk production and composition [11], abortion, and deaths in young animals, as well as the cost of diagnosis, treatment, and prevention. In European countries, losses have been estimated to exceed 20 million Euros/year [12,13], and depended on the herd size and lactation period [14].

*Mycoplasma agalactiae* (*Ma*) is reported to be the main causative pathogen for CA; however, *Mycoplasma capricolum* subsp. *capricolum*, *Mycoplasma mycoides* subsp. *capri,* and *Mycoplasma putrefaciens* have been found to cause a sporadic clinically similar syndrome, particularly in goats [5,15,16]. *Ma* is considered to be the main etiological agent of CA, which primarily affects sheep and goats along with other wild species. The main sources of *Ma* infection include milk, ocular, nasal, and auricular secretion, as well as through urine and feces [17]. The incubation period ranges from a few days to few weeks and even up to two months depending on the route of transmission, immune status of the host, and virulence of the infecting strain [17]. CA is certainly underreported and its prevalence varies hugely between regions across the world. No official data about CA incidence and prevalence are available [6].

Confirmation of CA can be obtained by isolating the causative mycoplasma. As isolation can take up to 2–3 weeks, PCR offers advantages in terms of time, specificity, and sensitivity. In addition, commercial ELISA kits have been described for serological detection [10].

In Sicily, Sardinia, and many Mediterranean basin countries, *M. agalactiae* is considered the dominant pathogen causing CA [4,18,19,20].

Control of CA relies on vaccines, treatment with anti-microbials, and appropriate herd management practices [6], but vaccination is the cornerstone in control and eradication of such disease [21].

In Italy, where CA is endemic, disease control is based on vaccination, anti-microbial treatment, and statutory measures, including quarantine of the infected herds. Unfortunately, antibiotics are not very effective because many of the anti-microbials used are often bacteriostatic rather than bactericidal; however, fluoroquinolones have been shown to be good in vitro, but are discouraged due to their high cost and the risk for the induction of resistance in bacterial pathogens into animal origin food chains [22]. Thus, vaccination remains the most cost-effective and reliable strategy.

Some assays were carried out about in preparation for safe and effective vaccines against *Ma* for the control of CA [23,24]. Live attenuated vaccines, used in Turkey, have been reported to be more efficient in the long term than inactivated vaccines [13,21,25], but should not be used during lactation and are not authorized in many European countries. The inactivated vaccine remains the most popular in Europe, although there is not a universally recognized one [10,19].

A combination of bacterial and/or viral vaccines containing multiple antigens has many benefits for manufacturers, as it reduces production costs; for administrators, as it saves time and efforts as well as simplifies the immunization schedule; and for the animals, as it minimizes the stress of multiple vaccinations [26].

The effectiveness of a vaccine to prevent disease depends on the potentiality/efficiency of the vaccine and is indicated by the development of a specific humoral immune response [27]. Seroconversion is useful to measure the induction of an immune response in the host and indicates the persistence of antibodies and immunity in the absence of disease [28,29].

The main scope of the present study is to try to answer a pressing question: does a bivalent vaccine elicit more immunity/protection against *Ma* than the traditional monovalent ones? The study is designed to assess whether, and to what extent and how, the combination of another pathogen (in our study *S. aureus*) in inactivated *Ma* vaccines could influence the developed humoral immunity, proportion of leukocyte subpopulations, hematological parameters, and SCC in lactating ewes.

## 2. Materials and Methods

### 2.1. Ethics Statement

All of the procedures involving animal were in accordance with the international guidelines for care and handling of experimental animals and were approved by the internal Ethic Committee for Animal Experiment in Istituto Zooprofilattico Sperimentale della Sicilia (Palermo, Italy), agreement no. 540/2019-PR (Resp. a Prot. 28875.31) issued on 07/23/2019, pursuant to Article 31 of Legislative Decree 26/2014. The methods were performed by approved staff members. The animals were handled with good clinical practices and efforts were made to minimize suffering.

### 2.2. Experimental Design

Twelve Valle del Belìce primiparous ewes, two years of age and in the same lactation stage, were selected from a *M. agalactiae*-free dairy herd composed of 200 heads reared in a semi-intensive production system. The *M. agalactiae* free status was verified by procedures suggested by the OIE manual [10]. The animals selected for the experiment were clinically examined and proven to be free from any mammary gland pathologies and were negative for anti-*M. agalactiae* antibodies using the ELISA test. Milk samples and nasal and auricular swabs from these animals were collected and analyzed by real-time PCR before starting the experiment. All of the samples were negative for *Ma* DNA. The animals were fed with a complete balanced diet and water ad libitum during the acclimatization and experimental periods, and were randomly divided into three groups in a separate paddock, each containing four ewes. First group was the unvaccinated control group and received the placebo (injected with 1 mL of physiological saline), the second group was subjected to the monovalent *M. agalactiae* inactivated vaccine, and the third group was injected with the *M. agalactiae–S. aureus (Ma–Sa)* bivalent (combined) inactivated vaccine. Animals were vaccinated through subcutaneous injection (1 mL/animal) with two doses of corresponding vaccine for each group at a 45-day interval (T0 and T1). As a result of unexpected snowfall (uncommon in the geographic area of the experimental farm) causing a drastic and sudden drop in temperature, the booster was administered after 45 days according to Nickerson et al. [30], in order to prevent a distorted immune response caused by intense cold stress from the animal. The first dose (T0) was administered 1.5 month after parturition.

The animals were observed and examined daily for clinical signs. Clinical scoring was based on the general behavior, food intake, abnormal local, and systemic reactions, including inflammation at the injection site and nasal discharge. The rectal temperature was recorded for each animal two days prior to vaccination and at the time of vaccination, and daily up to 14 days post vaccination (DPV). The site of vaccine inoculation was monitored for local adverse reactions. All of the animals were examined once a week for 17 successive weeks. Blood (two vacutainer tubes, each of 5 mL, one with ethylene diamine tetra acetic acid (EDTA) as an anti-coagulant and one without) and milk samples were collected weekly from all of the ewes.

### 2.3. Vaccines

The farm (autogenous) vaccine was produced by the vaccine laboratory at the Istituto Zooprofilattico Sperimentale della Sicilia in Palermo, Italy. The vaccine was prepared according to the guidelines governing the production of animal husbandry vaccines of veterinary interest (Ministerial Decree 17 March 1994, No. 287, GU General Series 11 of 14-05-1994). Both *Ma* and *Sa* were isolated from previous outbreaks in Sicily dairy sheep farms near this farm, and were confirmed by molecular identification.

For vaccine preparation, bacterial strains of *Ma* and *Sa* were grown in separate flasks.

Briefly, *Ma* was cultivated for 96 h at 37 °C in a microaerophile in Modified Hayflick Mycoplasma medium using PPLO broth base and was enriched with 20% of inactive equine serum [31].

*S. aureus* was cultured in TSB culture broth for 24 h at 37 °C under stirring.

Both bacterial cultures were centrifuged at 3800× *g* for 1 h and resuspended in PBS, pH 7.6, until reaching a titration of 10^9^ CFU/mL. The bacterial suspensions were then inactivated with 5% phenol for 24 h at 37 °C with stirring. The inactivated suspensions were adsorbed under constant and mechanical agitation for three hours at room temperature with aluminum hydroxide in a final concentration of 15%.

For the bivalent formulations, equal proportions of bacteria were added to constitute one volume (500 mL) of the bivalent vaccine at a final concentration of 10^9^ CFU/mL.

Sterility and safety tests were then carried out according to the current legislation (Ministerial Decree 17/3/94 N. 287). The vaccines were stored at 4 °C during the experimental period.

### 2.4. Serological Analysis

At the beginning of this study, the sera from all of the animals were collected from the jugular vein in a vacuum tube and were analyzed for the presence of specific antibodies against *Ma* with a commercial indirect enzyme-linked immunoassay (ELISA) method (CIVTEST Ovis *M. agalactiae* Laboratorios HIPRA, SA) [32]. The assay was done and evaluated for validity following the manufacturer’s guidelines. For interpretation of the results, an *Rz* value was calculated for each sample using the following Formula (1):(1)$$R_{z}=\frac{{\mathrm{OD}}_{450}\;\mathrm{Sample}}{2\times \left({\mathrm{Mean}\;\mathrm{OD}}_{450\;}\mathrm{Negative}\;\mathrm{Control}\right)}$$

(*Rz* ≤ 1.0) are negative samples, (1.5 ≤ *Rz* > 1.0) are suspicious samples, and (*Rz* > 1.5) are positive samples. Meanwhile, milk samples were proven to be negative for *M. agalactiae* by Rt-PCR.

Upon administration of the first dose of vaccine (T0 time) and for 17 successive weeks, the sera samples were collected from vaccinated and control sheep and analyzed for detection and the configuration of a concentration curve of anti-*Ma* antibodies.

### 2.5. Hematology Analysis

Blood samples, collected in vacutainer tubes containing EDTA, were rolled gently several times to ensure anticoagulant mixing, and were processed immediately for hematological analysis for complete blood counts with a differential cell count. Analysis was performed using the Abbott Cell-Dyn 3700 Hematology Analyzer using species-specific software for ovine samples and following the manufacturer instructions.

### 2.6. Somatic Cell Count (SCC)

Milk samples, preserved in 0.05% potassium dichromate, were transferred to the laboratory and analyzed within 24 h for SCC using an automated counter (Fossomatic FC 5000; Foss Electric, HillerØd, Denmark) [33].

### 2.7. Statistical Analysis

Data were presented as means ± standard deviation of the mean. The Kruskal–Wallis nonparametric one-way ANOVA was used to test the difference between groups, with a *p*-value < 0.05 considered significant. When significant, post hoc tests were performed using the Dunn nonparametric multiple comparisons test adjusted with Bonferroni correction using the FSA R (v 0.9.1.) package [34].

## 3. Results

No local or systemic reaction was observed in the animals injected during the study period.

Serological analysis for anti-*Ma* antibody detection by ELISA revealed a significant improvement in the humoral immune response in vaccinated animals. Here, all animals were free of antibodies against *M. agalactiae* at day 0 (the day of the first inoculation), and upon vaccination, the animals received both kinds of vaccines and showed seroconversion with increased antibody titer means. The level of Ab was increased after the first dose, and its peak occurred at around the 15th day then declined. After a booster dose on the 45th day (T1), Ab re-increased and reached its peak in both the monovalent and bi-valent vaccines at around 70th day before declining. Statistical analysis of the *Rz* value (representative of Ab titer) revealed a significant difference between the vaccinated and non-vaccinated animals (Table 1), as well as between the vaccinated groups (Table 2). The results indicated that both types of autogenous inactivated vaccines were potent and stimulated an immune response. The bivalent vaccine showed the highest significant level of stimulated Ab along the course of the study (Figure 1).

For hematological results, the total and differential leukocyte counts and hematological parameters were within the physiological range. Only RBCs and HCT showed slightly lower values than the normal physiological range in all of the groups. Statistical analysis of the data revealed a significant difference in the total leukocyte count between the vaccinated and non-vaccinated animals (*p*-value ≤ 0.05), with higher levels in the group that received the bivalent vaccine, although the difference between the vaccinated groups was not significant. The differential cell count showed a prominent increase in the neutrophil count in the animals that received the bivalent vaccine only, with a significant difference compared with the other two groups. There was no significant difference between the means of the lymphocytes, monocytes, and eosinophils among the three groups (Table 1; Figure 2). Hematological examination revealed a slight decrease in RBCs in the vaccinated animals compared with the control ones (*p*-value > 0.05). For Hb, there was slight decrease in the monovalent vaccinated animals (*p*-value > 0.05), while the means of the HTC did not show any significant difference among the three groups. Only the platelets showed an obvious and significant increase in the animals that received bi-valent vaccine compared with the other two groups (Table 1; Figure 3).

Concerning the SCC estimation throughout this study, we found that in the placebo injected control animals, SCC was relatively higher than for the other two groups, followed by a decline in SCC due to the dilution effect of increased productivity, with a re-increase towards the end of the study due to a decrease in milk production.

In the vaccinated animals, SCC was prominently lower than the placebo injected animals until the 6th week; however, it showed fluctuations but was always lower than for the control animals. This improvement in SCC was more prominent in animals vaccinated with a bi-valent vaccine than those that received a monovalent one; however, this difference was not significant (Table 1 and Figure 4).

## 4. Discussion

Vaccination is usually seen as an essential part of a comprehensive health program for both humans and animals in the fight against infectious diseases and in the prevention of future outbreaks [35]. In Italy, where CA is endemic, vaccination is the most cost-effective, sustainable, and reliable tool for the control of this disease [25]. One of the most common types of vaccines is autogenous “farm vaccines”, which are an important herd health tool, and are considered a feasible substitute to licensed vaccines [36]. In the case of CA, autogenous vaccines have gained much attention due to the absence of a universally commercial licensed vaccine [25]. In Sicily, an autogenous vaccine was prepared by the Istituto Zooprofilattico Sperimentale (IZS) using strains isolated from outbreaks, inactivated by formalin, and used under veterinary prescription [25].

In addition to quality antigens, effective vaccines need preferable adjuvants to enhance both cellular and humoral immunities. Moreover, the efficiency of inactivated vaccines is also potentially determined by the inactivation agent [37,38,39]. Mineral oil adjuvant-inactivated vaccines induce a higher and longer-lasting protective immunity, but can induce lesions at the injection site [24,40].

In the current study, Al-hydroxide was used as an adjuvant in both vaccines, because it is widely employed with efficiently established robust immune responses in veterinary medicine [41,42] and it is currently used in most commercial sheep vaccines [43].

Additional progress in vaccine production involved the preparation of combined vaccines that were produced to reduce production costs, save time and effort, simplify the immunization schedule, and minimize the stress of multiple vaccinations. Their efficacy has been proven in separate vaccines [44,45]. However, although numerous combined vaccines have been developed, there are relatively few studies describing the synergistic action of combination vaccines against sheep mastitis. This study tried to determine whether combining *S. aureus* in *M. agalactiae* inactivated vaccines could enforce the immune response and thus improve the efficacy of the vaccine.

The effectiveness/efficacy of a vaccine is determined either by serological analyses and/or challenge tests [27]. Seroconversion is useful to measure the induction of an immune response in the host and, in the absence of disease, it indicates the persistence of antibodies and immunity. Thus, seroconversion is a reliable indicator of vaccine efficacy [28,29].

For assessment of the humoral response/seroconversion, blood samples were collected before and after vaccination from animals from the three groups, and the elicited Ab was evaluated using the ELISA test, the recommended OIE test for the detection of seroconversion in CA [10]. The results of the study proved the development of a humoral immune response in both vaccines, but the bivalent combined vaccine showed a higher level of antibodies (Ab) along the course of the study (Figure 1).

In accordance with our findings, it was reported that antibody titers increased rapidly after vaccination with formalin-inactivated *Ma* vaccines and reached the highest level at 30 days before declining after 180 days [32,46]; however, others studies recorded the persistence of antibodies until 10 months after vaccination [47].

Our data are in accordance with the data reported on experimental vaccines combined with aluminum hydroxide, which are safe in sheep, but elicit low antibody titers that persist no longer than three months [24,46].

Our results show that both types of autogenous inactivated vaccines were potent to induce an immune response and that the bivalent one stimulated the highest level of antibodies.

However, the efficacy was evaluated shortly after booster vaccination and no data were provided about the ability of the vaccine to protect for longer periods.

The bivalent vaccine in the present study was proven to be safe, highly immunogenic, and fully protective.

On the basis of our findings, vaccination programs using this vaccine against CA should be planned to permit booster administration of the vaccine at least at four months intervals, providing an effective opportunity for the control of *Ma* infection in sheep herds in endemic areas. However, further investigation needs to be carried out.

Although the duration of immunity induced by vaccination is relatively short, the economic impact of CA in areas where small ruminant farming is prevalent should lead to a better understanding of the different aspects of this infection, and more attention should be paid to the development of a vaccine that is at the same time safe, effective, inexpensive, and easy to prepare, such as the one proposed in our work.

It is clearly demonstrated that combined (*Ma–Sa*) vaccines potentiated an immune response against *Ma* more than the *Ma* monovalent vaccines, with a similar antigenic concentration.

Other successful studies regarding combined mycoplasma vaccines have been reported. For instance, in cattle, the *M. mycoides* subsp. *Mycoides*/lumpy skin disease virus vaccine elicited a detectable antibody response similar to the monovalent vaccinations, with minimal adverse reactions [48]. In pigs, the *M. hyopneumoniae*/porcine reproductive and respiratory syndrome virus combined vaccine was more efficient than the single ones for controlling a dual Mhp/PRRSV infection [49]. The *M. hyopneumoniae*/Porcine CircoVirus type 2 combined vaccine was found to be safe and efficacious for controlling single and combined infections with these pathogens [50].

Similar results confirmed increased potentiality in combined vaccines of other pathogens in different animal species. In cattle, bovine respiratory syncytial virus/*Mannheimia haemolytica* combined vaccines were proven to have increased vaccine efficacy compared with single bovine respiratory syncytial virus vaccines [51,52]. In chickens, the combined inactivated *E. coli* and AI H5N1 vaccine generated a higher immune response and the protection rate was higher in the combined vaccine than the inactivated monovalent AI or *E. coli* vaccines [53]. Similarly, the combined AI and fowl cholera vaccine, and the *E. coli* and Newcastle inactivated vaccines showed a higher immune response than the monovalent ones [54,55]. On the other hand, other studies did not report a difference in the level of immune response upon combining peste des petit ruminants with sheep pox compared with single vaccines [56,57].

The main apprehension about combined vaccines is that an increasing number of antigens could theoretically pose problems in terms of reduced immunogenicity or increased reactogenicity [58,59]. In the current study, no adverse local or systemic reactions were detected and inoculated ewes produced high levels of specific antibodies starting from day 7 post-vaccination, suggesting that this combined vaccine is both safe and effective without any untoward reactions.

As complete blood count (CBC) usually complements the physical examination, it essentially supports diagnostic investigations and overall health management strategies in ruminants [60]. We looked for a deterioration in immune response in vaccinated animals through periodical hematological examination of the animals.

The total leukocytes and neutrophils were obviously higher in the vaccinated animals than the control ones, and the highest level was recorded in the animals that received the bi-valent vaccine. However, the total leukocyte count and distribution of granulocytes, monocytes, and lymphocytes remained within a normal range throughout the study in all of the animals. This indicated minimal systemic inflammatory responses to the multiple vaccines or to the method of administration. Only RBCs and HCT were slightly lower than the minimum physiological range in all of the groups, which might be attributed to the stress induced by first lactation or mineral deficiency.

Neutrophils, the most common polymorphonuclear leukocytes (PMNs) and found to be increased in this study, play a crucial role in innate immunity against infections [61]. It is now appreciated that PMNs can shape innate and adaptive immunity through regulating adaptive immunity and the induction of Ab production, but its role in vaccine-induced protection in vivo remains unclear [62]. Neutrophils were viewed as effectors of vaccine responses, as vaccination triggers the Abs that bind to pathogens and promote their clearance via enhancing uptake and killing by PMNs [63]. A recent study recorded that neutrophils were needed for the production of functional Abs following vaccination, as well as at the time of immunization for full protection against subsequent invasive infection. The authors concluded that neutrophils were required during vaccination for optimal host protection against pneumococci and other pathogens [64].

Platelets were prominently increased in the animals that received a bi-valent vaccine compared with the other two groups in the current study. There is an increasing focus on platelets as key regulators of the immune system. Platelets interact directly with pathogen-associated molecular patterns (PAMPs) [65,66]. They possess a suite of pattern recognition receptors (PRRs) and facilitate the recruitment of other innate immune effectors such as granulocytes, monocytes, and innate lymphoid cells, including natural killer cells [67]. Moreover, stimulated platelets interact with neutrophils and lead to the formation of neutrophil extracellular traps, which capture circulating bacteria [68].

Recently, platelets were reported to play a key role in the coordination of all components of the innate immune system, producing an initial response in their own right and interacting with myeloid, lymphoid, and humoral components of the initial response to infection [69].

Some pathogens have been shown to interact with platelets directly, such as *Streptococcus pyogenes*, *S. aureus,* Candida species, and Aspergillus spp., through specific virulence factors that target platelets [70,71,72,73,74]. These virulence factors may have evolved as a means of pathogen for circumvention of the immune response, facilitated by platelets.

Finally, as SCC is considered from many aspects as an indicator of udder health [75,76,77], we measured the SCC weekly along the course of the study. We found that SCC was lower in the vaccinated animals, particularly for the bi-valent one, than for the control animals during the first six weeks (Figure 4), then increased towards the end of study. The SCC profile in the control animals is matched with previous studies and represents a physiological response with a progression in lactation [78,79]. The current results indicate an improvement and maintenance of immune and health status of the udder upon vaccination. Our findings are in accordance with previously reported results. For instance, in dairy goats vaccinated against staphylococcal mastitis using bacteria, the authors recorded a decrease in the average SCC of the milk samples from vaccinated animals compared with the controls [80]. In cattle, vaccination with the *Salmonella* Newport SRP vaccine significantly decreased SCC at 30 to 60 days of lactation [81]. Other studies evaluated the direct effects of the inactivated *S. aureus* vaccine in dairy Buffalo using different protocols. The authors recorded lowering in SCC in vaccinated animals compared with the control non-vaccinated animals [82,83]. On the other hand, some authors have demonstrated that vaccination did not have any effect on SCC [84,85,86].

The findings of this study indicate that combining *S. aureus* with *M. agalactiae* enhances immune stimulation and prove that the *Ma–Sa* combined vaccine is more potent for stimulating the immune system than a separate Ma vaccine. The reason for this is that it still needs to be identified whether this was due to more antigenic stimuli or directly attributed to *S. aureus,* as some previous literature reported that *S. aureus* harbors virulence factors that target and activate platelets. Even though this study included a small number of animals, its findings about the potentialities of this inactivated vaccine in the control of CA are strongly encouraging. This might need further confirmation through additional replicates and further study is needed before proceeding with widespread use.

## 5. Conclusions

The current study represents the first investigation evaluating the synergistic effect of combining *S. aureus* in an inactivated *M. agalactiae* vaccine and its impact on stimulating elicited Ab. The use of an inactivated combined vaccine was associated with a more enforced immune response in animals compared with the single inactivated *M. agalactiae* (monovalent) vaccine. The results proved a significant improvement in humoral response and hematological parameters, manifested by an increased Ab titer, total leukocyte count, neutrophils, and blood platelets. This might be attributed to the activation effect of *S. aureus* on platelets, which is recoded to act as a key regulator for the coordination of all components of the innate immune system, producing an initial response and interacting with other components of the initial response to infection. Moreover, the local glandular immunity of the udder, assessed by SCC, was enhanced.

## Figures and Tables

**Figure 1 vetsci-09-00651-f001:**
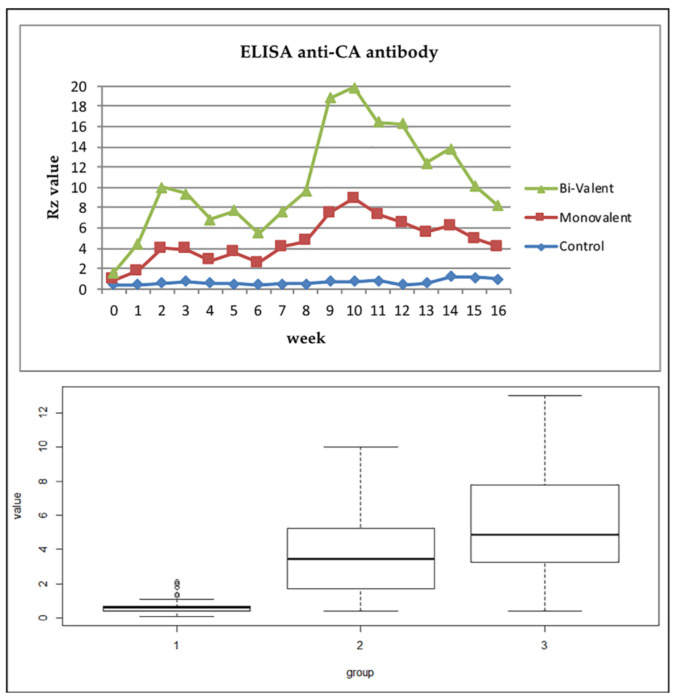
Results of the estimation of the anti-CA antibodies in the serum using ELISA (graphs and boxplots). *Rz* value: Proportional to the amount of Ab according to the ELISA manual; group 1 (control), group 2 (monovalent vaccine), group 3 (bi-valent vaccine).

**Figure 2 vetsci-09-00651-f002:**
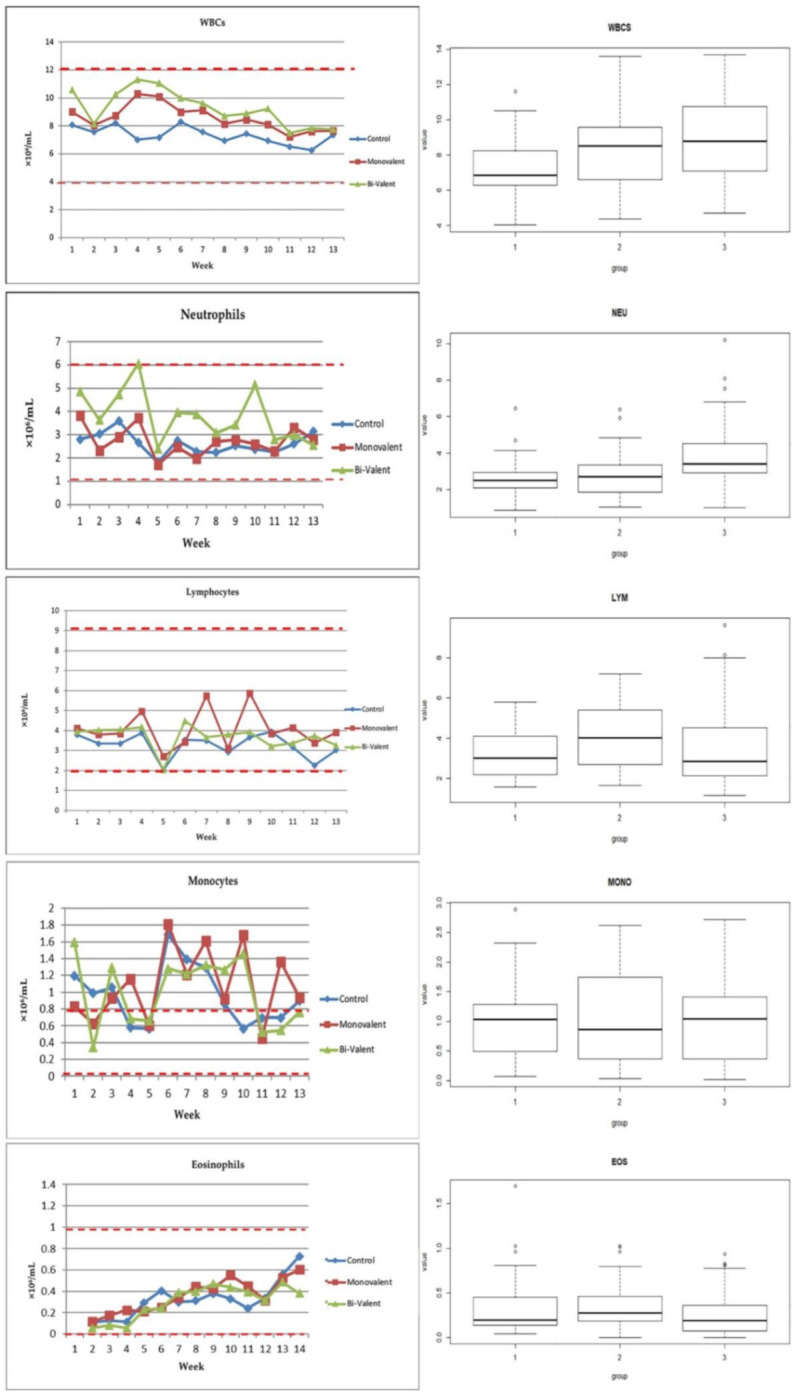
Total and differential leukocyte count (graphs and boxplots). WBCs: total leukocytic count; NEU: neutrophils; LYM: lymphocytes; MONO: monocytes; EOS: eosinophils; group 1 (control), group 2 (monovalent vaccine), group 3 (bi-valent vaccine); the red dotted line refers to the physiological threshold of the parameter value in ovine species.

**Figure 3 vetsci-09-00651-f003:**
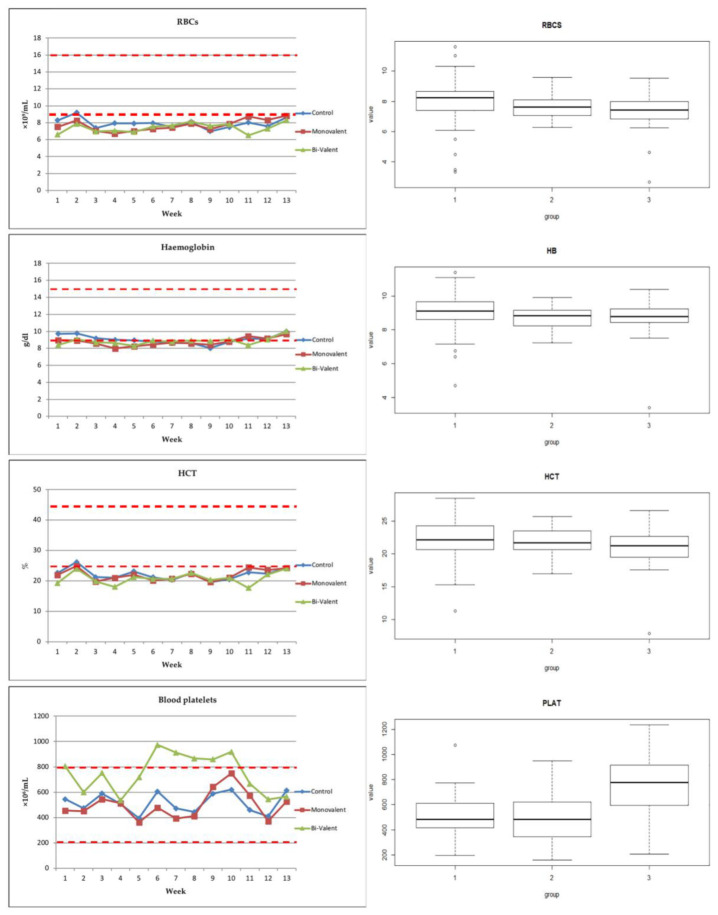
Results of hematological examination (graphs and boxplots). RBCs: red blood cells; HB: hemoglobin; HCT: hematocrit value; PLAT: platelets; group 1 (control), group 2 (monovalent vaccine), group 3 (bi-valent vaccine). The red dotted line refers to the physiological threshold of the parameter value in ovine species.

**Figure 4 vetsci-09-00651-f004:**
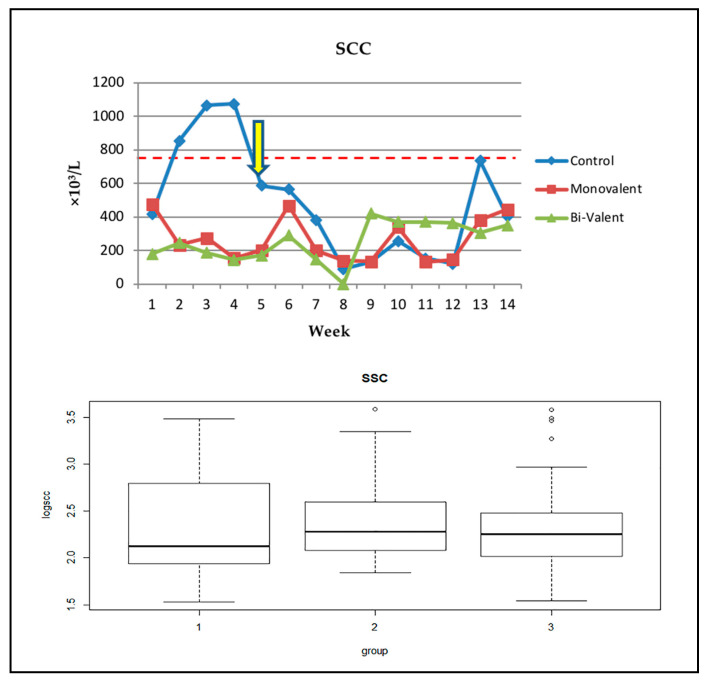
Results of SCC (graphs and boxplots). SCC: somatic cell count; group 1 (control), group 2 (monovalent vaccine), group 3 (bi-valent vaccine). The red dotted line refers to the physiological threshold of the parameter value in ovine species while the yellow arrow indicates the first decrease in SCC after the placebo administration below physiological threshold.

**Table 1 vetsci-09-00651-t001:** Results of the statistical analysis of the estimated variables (Ab titer, hematological parameters, and SCC).

Variable	Control Group Mean ± SD	Mono-Valent Group Mean ± SD	Bi-Valent Group Mean ± SD	Kruskal-Wallis H (*p*-Value)
*Rz* value (Ab titer)	0.67 ± 0.37	3.8 ± 2.60	5.54 ± 3.21	**2.2 × 10^−16^**
WBCs (×10^6^/mL)	7.2 ± 1.54	8.35± 2.10	3.87 ± 1.76	**3.1 × 10^−4^**
NEU (×10^6^/mL)	2.62 ± 0.91	2.77 ± 1.23	3.87 ± 1.76	**3.9 × 10^−5^**
LYM (×10^6^/mL)	3.24 ± 1.22	4.05 ± 1.59	3.66 ± 2.18	**0.053**
MONO (×10^6^/mL)	0.97 ± 0.60	1.07 ± 0.76	0.99 ± 0.67	**0.834**
EOS (×10^6^/mL)	0.33 ± 0.30	0.34 ± 0.26	0.28 ± 0.26	**0.250**
RBCs (×10^9^/mL)	7.96 ± 1.52	7.64 ± 0.80	7.37 ± 1.16	**8.2 × 10^−3^**
HB (g/dL)	9.05 ± 1.14	8.73 ± 0.65	8.75 ± 1.03	**0.038**
HCT (%)	22.2 ± 3.18	21.8 ± 2.21	21.06 ± 2.88	**0.12**
PLAT (×10^6^/mL)	517.12 ± 156.46	495.02 ± 198.62	739.39 ± 219.00	**6.39 × 10^−8^**
SCC	1795 ± 4279	1085 ± 2349	1920 ± 5308	**0.697**

*Rz* value: Proportional to the amount of Ab according to the ELISA manual; WBCs: total leukocytic count; NEU: neutrophils; LYM: lymphocytes; MONO: monocytes; EOS: eosinophils; RBCs: red blood cells; HB: hemoglobin; HCT: hematocryte value; PLAT: platelets; SCC: somatic cell count.

**Table 2 vetsci-09-00651-t002:** Dunn post hoc test of significantly different variables.

Variable	Group	Z-Test-Statistics	*p*-Adjusted
*Rz* value	Control-Mono	−7.21	1.64 × 10^−12^
	Control-Bivalent	−9.70	8.74 × 10^−22^
	Mono-Bivalent	−2.41	0.047
WBCS	Control-Mono	−2.91	0.01
	Control-Bivalent	−3.8	0.00
	Mono-Bivalent	−0.83	1.00
NEU	Control-Mono	−0.59	1.00
	Control-Bivalent	−4.24	6.68 × 10^−5^
	Mono-Bivalent	−3.49	1.42× 10^−3^
RBCs	Control-Mono	2.2	0.08
	Control-Bivalent	2.95	0.01
	Mono-Bivalent	0.7	1.00
HB	Control-Mono	2.43	0.046
	Control-Bivalent	1.82	0.21
	Mono-Bivalent	−0.6	1.00
PLAT	Control-Mono	0.37	1.00
	Control-Bivalent	−4.95	2.27 × 10^−6^
	Mono-Bivalent	−5.08	1.14 × 10^−6^

*Rz* value: Proportional to the amount of Ab according to the ELISA manual; WBCs: total leukocytic count; NEU: neutrophils; RBCs: red blood cells; HB: hemoglobin; PLAT: platelets.

## Data Availability

Data are contained within the article.

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
