# Peer review of "Preliminary Study on the Host Response to Bivalent and Monovalent Autogenous Vaccines against Mycoplasma agalactiae in Dairy Sheep"

_vetsci, 2022, doi:10.3390/vetsci9120651_

Round 1

Reviewer 1 Report

Well written - comprehensive and clear.

As authors claim an innovative approach not reported before.

Comprehensively referenced

Design - some concerns over sample  size - why not larger groups - since 200 herd.  I'm not expert statistician - but recall group min. group-size of 5 needed for K-W?

how was herd selected - any tests in animals in the herd but not in study?

Figure quality should be improved - reduce font title & size y-axes , and add label x-axes (hence lower scores above)

L220 remove 'that'

L238/239 'until the'

L241 'which received'

L298 - quality antigens

L332 lumpy

Author Response

We would like to thank you for your time and efforts in handling our manuscript. We are grateful to the referees for their critical evaluation of our manuscript and for their helpful questions, suggestions and corrections. Please find below our replies item by item (red font) to each of the statements/queries raised by the referees

Design - some concerns over sample size - why not larger groups - since 200 herd. I'm not expert statistician - but recall group min. group-size of 5 needed for K-W?

The Kluskal Wallis is a non-parametric method that combine the observations in the k samples into a single pooled 'null' sample. Within each group we have more than 5 observations then the assumptions of the Kruskal-Wallis test are respected.

How was herd selected - any tests in animals in the herd but not in study?

Thank you for the observation. We added the missing information specifying that the preliminary investigations were carried out throughout the herd and not only in selected animals

Figure quality should be improved - reduce font title & size y-axes , and add label x-axes (hence lower scores above)

Thank you for the suggestion. We changed all pictures improving their resolution and presentation

L220 remove 'that'

L238/239 'until the'

L241 'which received'

L298 - quality antigens

L332 lumpy

Thank you for the correction. We corrected all typos

Reviewer 2 Report

Your research article about the host response to monovalent and bivalent autogenous vaccine against MA in dairy sheep is important and relevant.

I have following suggestions and comments for you to consider:

Introduction:

To provide clearer context please include more epidemiology about ovine contagious agalactia, including: how is disease transmitted, what are susceptible species, what is incubation period, what are incidence and prevalence in Italy/Europe, how is it usually diagnosed. 

Line 75: include in what locations abscesses can be caused by MA.

 Materials and Methods: 

line127: please specify how the 12 ewes were selected.

lines 138-140: you should state where the chosen vaccination site/s were. What was the reason for having a 45- day interval between vaccinations? This should be stated.

You should also state where the ewes were kept during the experiment.

2.4 Serological analysis: for completeness, please describe the method and location of venipuncture for collection of blood samples.

Why was a SA group only vaccine not included? I think that should be briefly discussed.

Results:

In figure 1 -3 please provide units for the x-axis. And also include when vaccine/placebo were given. 

Discussion: good logical discussion. You could consider to include a brief discussion about your observation of the length of raised antibody levels. Is this long enough? What are the practical implications of relative short duration?

Overall, please review the English-there are some grammatical and spelling mistakes throughout the manuscript.

Author Response

We would like to thank you for your time and efforts in handling our manuscript. We are grateful to the referees for their critical evaluation of our manuscript and for their helpful questions, suggestions and corrections. Please find below our replies item by item (red font) to each of the statements/queries raised by the referees

To provide clearer context please include more epidemiology about ovine contagious agalactia, including: how is disease transmitted, what are susceptible species, what is incubation period, what are incidence and prevalence in Italy/Europe, how is it usually diagnosed. 

Thank you for the suggestion. We added all the information requested (lines 84-94)

Line 75: include in what locations abscesses can be caused by MA.

Thank you for the suggestion. We added the missing information

line127: please specify how the 12 ewes were selected.

Thank you for the suggestion. We added the missing information

lines 138-140: you should state where the chosen vaccination site/s were. What was the reason for having a 45- day interval between vaccinations? This should be stated.

Thank you for the suggestion. We added the requested information (lines 172-175)

You should also state where the ewes were kept during the experiment.

Thank you for the suggestion. We added the missing information

2.4 Serological analysis: for completeness, please describe the method and location of venipuncture for collection of blood samples.

Thank you for the suggestion. We added the missing information

Why was a SA group only vaccine not included? I think that should be briefly discussed.

Thanks for the comment. The aim of the work was to create a more efficient vaccine against M. agalactiae, therefore it was not considered appropriate to evaluate a vaccine consisting only of S. aureus

In figure 1 -3 please provide units for the x-axis. And also include when vaccine/placebo were given. 

Thank you for the suggestion. We changed all pictures improving their resolution and presentation

Discussion: good logical discussion. You could consider to include a brief discussion about your observation of the length of raised antibody levels. Is this long enough? What are the practical implications of relative short duration?

 Thank you for the suggestion. We added a discussion about our observation on the length of raised antibody levels (lines 409-426)